# Infraslow Neurofeedback Training Alters Effective Connectivity in Individuals with Chronic Low Back Pain: A Secondary Analysis of a Pilot Randomized Placebo-Controlled Study

**DOI:** 10.3390/brainsci12111514

**Published:** 2022-11-08

**Authors:** Divya Bharatkumar Adhia, Ramakrishnan Mani, Paul R. Turner, Sven Vanneste, Dirk De Ridder

**Affiliations:** 1Department of Surgical Sciences, Otago Medical School, University of Otago, Dunedin 9016, New Zealand; 2Pain@Otago Research Theme, University of Otago, Dunedin 9016, New Zealand; 3Department of Physiotherapy, Manipal College of Health Professionals, Manipal Academy of Higher Education, Manipal 576104, India; 4Centre for Health, Activity and Rehabilitation Research, School of Physiotherapy, University of Otago, Dunedin 9016, New Zealand; 5Global Brain Health Institute, Trinity College Dublin, 2 Dublin, Ireland

**Keywords:** neurofeedback, brain computer interface, chronic low back pain, randomized controlled trial

## Abstract

This study explored the effect of electroencephalographic infraslow neurofeedback (EEG ISF-NF) training on effective connectivity and tested whether such effective connectivity changes are correlated with changes in pain and disability in people with chronic low back pain. This involved secondary analysis of a pilot double-blinded randomised placebo-controlled study. Participants (*n* = 60) were randomised to receive ISF-NF targeting either the pregenual anterior cingulate cortex (pgACC), dorsal anterior cingulate and somatosensory cortex (dACC + S1), ratio of pgACC*2/dACC + S1, or Sham-NF. Resting-state EEG and clinical outcomes were assessed at baseline, immediately after intervention, and at one-week and one-month follow-up. Kruskal–Wallis tests demonstrated significant between-group differences in effective connectivity from pgACC to S1L at one-month follow up and marginal significant changes from S1L to pgACC at one-week and one-month follow up. Mann–Whitney U tests demonstrated significant increases in effective connectivity in the ISF-NF up-training pgACC group when compared to the Sham-NF group (pgACC to S1L at one-month (*p* = 0.013), and S1L to pgACC at one-week (*p* = 0.008) and one-month follow up (*p* = 0.016)). Correlational analyses demonstrated a significant negative correlation (ρ = −0.630, *p* = 0.038) between effective connectivity changes from pgACC to S1L and changes in pain severity at one-month follow-up. The ISF-NF training pgACC can reduce pain via influencing effective connectivity between pgACC and S1L.

## 1. Introduction

Electroencephalography (EEG)-based neurofeedback (NF) training is a non-invasive brain–computer interface biofeedback technique that facilitates an individual’s ability to self-regulate their real-time cortical activity in the targeted brain regions and reinforce learning through operant conditioning [1]. In other words, individuals can modify the electrical activity of the targeted brain region in the desired direction through a closed-loop feedback system, in which an exogenous sensory stimulus (e.g., auditory tone) is fed back to the individuals in real-time following the attainment of the desired neural activity. Thus, with the possibility of endogenously manipulating brain activity as an independent variable, EEG-NF is a powerful neuroscientific tool that has been explored for the treatment of several clinical conditions with altered cortical activity [1].

With considerable neuroimaging evidence of cortical activity alterations in individuals with chronic pain [2,3,4,5,6,7,8], closed-loop NF training has been explored by several non-randomised studies and pilot randomised controlled trials as a potential treatment option for chronic pain conditions (e.g., fibromyalgia, headache, neuropathic pain) [9,10,11]. Recent systematic reviews and meta-analyses highlight that, although the confidence is low, the EEG-NF may offer clinically meaningful benefits in short-term pain intensity and pain interference in people with chronic pain [9,10,11]. Furthermore, a few studies also demonstrate improvements in pain-associated symptoms (such as depression, anxiety, fatigue, and sleep) [12,13,14], which are well-known to have a detrimental effect on ongoing pain, contributing to disability and poor quality of life. EEG-NF could thus be one of the powerful holistic treatment approaches for improving overall well-being in people with chronic pain and requires further investigation.

While the evidence for the effect of EEG-NF looks promising, it is not without criticisms. One of the major criticisms, based on multiple trials in people with attention deficit hyperactivity disorder, is the lack of evidence of the specificity of treatment effects [15]. Some authors claim that the EEG-NF benefits participants regardless of the feedback source [16,17,18,19]. A few studies demonstrate that the sham neurofeedback of an unrelated signal exhibits an equivalent treatment effect as the real NF [16,17,18,19]. Moreover, very few studies assess and/or demonstrate evidence of objective neural changes associated with improvements in clinical outcomes. These factors have led sceptics to attribute most relevant experimental findings and changes in clinical outcomes to placebo factors [15,20]. Thus, to determine the specificity of the EEG-NF training, it is of utmost importance to evaluate whether EEG-NF training influences neural changes and to test whether such changes in the neural activities post-training are associated with changes in clinical outcomes.

EEG-NF is believed to act potentially by changing the connectivity to/from the trained area [1]. Effective connectivity is a measure inferring the directional functional connectivity between brain regions [21,22]. Granger causality reflects the strength of effective connectivity (i.e., causal interactions) from one region to another by quantifying how much the signal in the seed region can predict the signal in the target region [21,22]. In other words, Granger causality can be considered as a directional functional connectivity. As EEG-NF training has the potential to influence the effective connectivity between cortical areas of interest, the aim of this study was to explore the effect of EEG ISF-NF training on effective connectivity between the targeted brain regions (pgACC, dACC, and S1), and to determine its associations with the changes in the pain and disability.

## 2. Materials and Methods

### 2.1. Trial Registration and Ethical Approval

Prospectively registered in the Australian and New Zealand Clinical Trials Registry (https://www.anzctr.org.au/Trial/Registration/TrialReview.aspx?id=379470&isReview=true accessed on 11 November 2021).; registration number: ACTRN12620000414910; date of registration: 27 March 2020), this study was conducted according to the ethical standards of the 1964 Declaration of Helsinki. The NZ Health and Disability Ethics Committee provided ethical approval for this study (Ref:20/CEN/60). All participants provided written informed consent prior to study enrolment.

### 2.2. Study Design

This pilot study was a double-blinded randomized placebo-controlled parallel trial with four intervention arms, with the clinical and EEG measures collected at baseline (T0), and immediately (T1), one week (T2), and one month (T3) post-intervention

This investigation was a secondary analysis of a pilot randomised placebo-controlled study conducted to explore the feasibility and safety of a novel source-localized EEG-NF for the treatment of chronic low back pain (CLBP), targeting the infraslow frequency (ISF) electrical activity in the three cortical areas involved in pain processing, namely the dorsal anterior cingulate cortex (dACC), primary somatosensory cortex (S1), and the pregenual anterior cingulate cortex (pgACC) [2,3,23,24,25]. The dACC encodes pain unpleasantness [26,27,28,29,30]; in other words, the unpleasant emotional component of pain. The S1 processes the discriminatory/sensory components of the pain, such as pain intensity, pain localization, and pain character (burning, aching, etc.) [28,31,32]. The dACC and the S1 are parts of the two ascending pain pathways: namely medial and lateral, respectively. The two ascending pain pathways are balanced by a descending pain inhibitory pathway [33,34], involving the pgACC, the periaqueductal gray, the para-hippocampal area, the hypothalamus, and the rostral ventromedial brainstem [33,34,35]. The primary pilot study (under review) was performed to train the brain via brain–computer interface training to normalize its balance through three approaches: 1. strengthening pain inhibition via up-training pgACC activity; 2. reducing pain-provoking activity via down-training dACC and S1; and 3. Restoring the balance via simultaneously up-training pain inhibition and down-training pain-provoking activity. These three approaches were compared to the placebo-group in which training was non-specific.

#### 2.2.1. Randomization

A research administrator, not involved in treatment/assessment, randomized, and assigned participants using computerized open-access randomization software program, without applying any restrictions (on a 1:1:1:1 basis) to either:Group 1: ISF-NF up-training pgACC, i.e., modulate the descending pain inhibitory pathwayGroup 2: ISF-NF down-training dACC + S1, i.e., modulate the medial and lateral ascending pain pathwayGroup 3: ISF-NF concurrently up-training pgACC and down-training dACC + S1, i.e., the ratio of [(2xpgACC)☹dACC + S1], i.e., normalize the balance between the descending inhibitory and ascending pain pathwaysGroup 4: Sham-NF

The randomisation schedule was concealed in sequentially numbered sealed opaque envelopes and provided to participants at baseline.

#### 2.2.2. Blinding

Participants and outcome assessor were blinded. The success of blinding was assessed after the completion of the intervention using the question “What type of treatment do you believe that you/the participant received respectively?” The confidence in their judgement was assessed on a 11-point NRS (0 = Not at all to 10 = Extremely confident), with the reason being noted, and whether the intervention was revealed to them.

### 2.3. Participants and Eligibility Criteria

Sixty participants were enrolled and randomised equally into four treatment groups (Figure 1). Of the total participants enrolled (*n* = 60), we lost 8 participants following the baseline assessment session (Figure 1). The most common reason for dropouts was the time commitment required and fitting treatment sessions around the participant’s work schedule. Furthermore, seven participants discontinued treatment due to various reasons (outlined in Figure 1). Thus, the number of participants that completed all the neurofeedback treatment sessions was 12 in Group 1 and 11 each in Groups 2 to 4.

Interested volunteers were screened for eligibility and enrolled by a researcher with an advanced qualification in musculoskeletal physiotherapy. Adults between the ages of 18 and 75 years who had pain in the lower back region for ≥3 months, had a score of ≥4 on an 11-point Numeric pain rating scale (NPRS) [36] in four weeks prior to enrolment, and had a disability score of ≥5 on the Roland–Morris Disability Questionnaire (RMDQ) [37] were eligible to participate in the study. Volunteers with the following conditions/situations were excluded: inflammatory arthritis, auto-immune conditions, undergoing physiotherapy/chiropractic therapy, recent back injuries in the last 3 months, radicular pain/radiculopathy, spinal surgery/lumbar epidural injections in the last 6 months, current intake of centrally acting medications or intention of taking new medications in the next three months, neurological diseases, substance abuse, dyslipidaemia, unstable medical/psychiatric conditions, epilepsy/seizures, peripheral neuropathy, vascular disorders, cognitive impairments, hearing problems, recent/current pregnancy, and presence of any electronic implants.

At baseline assessment, all participants completed questionnaires to capture demographics, clinical characteristics of CLBP, including the presence of central sensitivity (Central Sensitization Inventory) [38], neuropathic pain (PainDETECT) [39], treatment expectancy/credibility [40], sleep (Medical Outcomes Study-Sleep Scale) [41], psychological measures (Depression, Anxiety, Stress Scale [42], Pain Catastrophizing Scale [43], Pain Vigilance Awareness Questionnaire [44], Positive and Negative Affect Schedule-short form [45], Emotion Regulation Questionnaire [46], and Five-Facet Mindfulness Questionnaire-15 [47]), and general well-being (European Quality of Life [48] and WHO-Five Well-Being Index [49]).

Table 1 presents descriptive data of the participants at baseline, indicating that all groups were comparable.

### 2.4. Intervention

Source-localised EEG ISF-NF [50] was administered three times a week (30 min/session) for four consecutive weeks (12 sessions) by a researcher (with physiotherapy background) experienced in delivering neuromodulation techniques. Treatment was delivered using a 21-channel DC coupled amplifier and BrainAvatar software version 4.7.5 produced by Brainmaster Inc., Bedford, OH, USA. The sLORETA source localization permits the selection of any region of brain for feedback of current density, using voxels as regions of interest (ROI), which are selected based on MNI coordinates. The current densities for the chosen voxels are computed continuously using fast Fourier transformation and inverse solution sLORETA software for the targeted brain regions and can be fed back to participants by using sound feedback.

During each session, the Comby EEG lead cap with 19 (Ag/AgCl) electrodes positioned according to International 10–20 system was fixed to the individual’s scalp. The impedance of electrodes was monitored and kept below 5 kilo-ohms. Participants were instructed to close their eyes, relax, minimize their movements, and listen to the sound feedback. The system delivered sound feedback (reward) each time the participant’s brain activity met the desired infraslow (0.0–0.1 Hz) threshold in targeted brain regions. No explicit instructions regarding mental strategies to be used during NF training were provided.

#### 2.4.1. ISF-NF Treatment Groups

For the current study, we developed EEG-NF training programs to up-train (i.e., increase current density) ISF activity at pgACC (Group 1) and down-train (i.e., decrease current density) ISF activity simultaneously at dACC and S1 (Group 2). For Group 3, a program was developed to concurrently up-train ISF activity at pgACC (x2) and down-train ISF activity at dACC + S1, in order to reinforce the ratio between these regions as being >1, as below:(1)$$\mathrm{Ratio}=\frac{2\;x\;\mathrm{pgACC}}{\mathrm{dACC}+S1}>1$$

For all groups, the reward threshold was adjusted in real-time between 60 and 80%, i.e., for 60–80% of the time, sound feedback was delivered by the system when participant’s brain activity met the desired infraslow magnitude.

#### 2.4.2. Sham-NF Group

To create identical auditory feedback to the ISF-NF groups, participants in the sham-NF group listened to a random set of pre-recorded sound files (*n* = 12), sourced from a database of recorded audio files (using audacity software version 3.2.1) of healthy participants that underwent EEG source-localised ISF-NF training (targeting ratio between pgACC and dACC + S1). All other conditions were kept same as in the ISF-NF groups.

### 2.5. Outcome Assessment

#### 2.5.1. Electroenecephalography

Resting-state eyes-closed EEG (~10 min) was performed using SynAmps-RT Amplifier (Compumedics-Neuroscan). Sixty-four electrodes were placed in 10–10 International placement and impedances were checked to remain below 5 kΩ. Data were resampled (128 Hz), band-pass filtered (0.005–0.2 Hz), plotted in EEGLAB and IcoN software version 3 for careful inspection and manual artefact rejection.

Effective connectivity analyses: Granger causality reflects the strength of effective connectivity (i.e., causal interactions) from one region to another by quantifying how much the signal in the seed region can predict the signal in the target region [21,22]. In other words, Granger causality can be considered as a directional functional connectivity. It is based on formulating a multivariate autoregressive model and calculating the corresponding partial coherences after setting all irrelevant connections to zero [51]. In general, the autoregressive coefficients correspond to Granger causality [22,52]. It is defined as the log-ratio between the error variance of a reduced model, which predicts one time series based only on its own past values, and that of the full model, which in addition, includes the past values of another time series. It is important to note that Granger causality does not imply anatomical connectivity between regions, but rather directional functional connectivity between two sources.

Standardized low-resolution brain electromagnetic tomography (sLORETA) source localisation software was used to estimate for effective connectivity analyses. The Granger causality values were calculated as a measure of effective connectivity between the three targeted ROIs (pgACC, dACC, and left and right S1) for three ISF bands: ISF1 (low: 0.01–0.04 Hz), ISF2 (mid: 0.05–0.07) and ISF3 (high: 0.08–0.10).

#### 2.5.2. Clinical Measures

Pain intensity and interference: the Brief Pain Inventory (BPI) [53], a standardised, validated questionnaire, was used to capture pain intensity and interference in daily activities. Higher scores indicate higher levels of severity and interference, respectively.Pain unpleasantness and bothersomeness in the last 24 h were measured using NRS (0 = not unpleasant/bothersome to 10 = most unpleasant/bothersome imaginable) [54,55]. Higher scores indicate higher levels of unpleasantness/bothersomeness, respectively.Physical Function: RMDQ [37], a 24-item questionnaire widely used in clinical research with proven validity and reliability in individuals with CLBP, was applied to assess self-reported functional abilities. RMDQ was used as a measure to evaluate the effect of ISF-NF on disability. Higher scores indicate higher levels of disability.

### 2.6. Statistical Analysis

Data were analysed using SPSS_v27.0. We conducted these secondary analyses only on the 45 participants that completed all the training sessions.

Mean differences were calculated for all the EEGs and the clinical measures from baseline (T0) to each interim (T1, T2, and T3). As this was a pilot exploratory study with a small sample size, we used non-parametric tests to compare the between-group differences at each time-point (T1, T2, and T3).

A stepwise approach was followed for the analyses. First, a Kruskal–Wallis H test (K-Independent samples) was used to determine the significant group-level differences in the effective connectivity measures. A *p* value of ≤0.05 was considered significant. Individual tests were conducted for each effective connectivity measure in each of the three frequency bands at each time-point (T1, T2, and T3).

Second, for all the effective connectivity measures that demonstrated a significant difference in the Kruskal–Wallis H test, we then performed a Mann–Whitney U test (two independent samples) to individually compare each of the three treatment groups (i.e., pgACC, dACC + SSC, and ratio) to the sham-NF treatment group. To adjust for multiple comparisons, a *p* value of ≤0.017 was considered significant.

Third, a bivariate correlational analysis was conducted for the effective connectivity measures that demonstrated significant differences in the treatment group compared to the sham group. We used Spearman’s correlational analysis to determine the associations between the changes in the effective connectivity measures and the changes in the clinical outcomes of pain and disability.

## 3. Results

### 3.1. Effective Connectivity Measures

#### 3.1.1. ISF1 (Low: 0.01–0.04 Hz)

The findings for the effective connectivity analysis in the ISF1 are presented in Table 2 and Figure 2. The Kruskal–Wallis H test demonstrated a significant group-level difference in the effective connectivity from pgACC to S1L at one-month (T3) follow up, and a marginal significant group-level difference from S1L to pgACC at one-week (T2) and one-month (T3) follow up.

The Mann–Whitney U tests were conducted for these effective connectivity measures to individually compare the ISF-NF treatment groups with the Sham-NF group.

Effective connectivity in the direction of pgACC to S1L at one-month follow-up (T3): A significant increase was noted post ISF-NF up-training pgACC (Z = −2.22, *p* = 0.013) when compared to the Sham-NF group (Figure 2). No significant differences were noted post ISF-NF down-training dACC + S1 (Z = 0.00, *p* = 0.514) or training ratio (Z = −0.43, *p* = 0.350) when compared to the Sham-NF.

Effective connectivity in the direction of S1L to pgACC at one-week follow-up (T2): A significant increase was noted post ISF-NF up-training pgACC (Z = −2.4, *p* = 0.008) when compared to the Sham-NF group (Figure 2). No significant differences were noted post ISF-NF down-training dACC + S1 (Z = −1.59, *p* = 0.059) or training ratio (Z = −0.69, *p* = 0.519) when compared to the Sham-NF.

Effective connectivity in the direction of S1L to pgACC at one-month follow-up (T3): A significant increase was noted post ISF-NF up-training pgACC (Z = −2.15, *p* = 0.016) when compared to the Sham-NF group (Figure 2). No significant differences were noted post ISF-NF down-training dACC + S1 (Z = −0.07, *p* = 0.486) or training ratio (Z = −0.62, *p* = 0.281) when compared to the Sham-NF.

#### 3.1.2. ISF2 (Mid: 0.05–0.07)

The findings for the effective connectivity analysis in the ISF2 are presented in Table 2 and Figure 3. The Kruskal–Wallis H test demonstrated no significant group-level differences in the effective connectivity measures in the ISF2 band at all the three time points (T1, T2, T3). Thus, no Mann–Whitney U tests to individually compare the active treatment groups to sham group were conducted.

#### 3.1.3. ISF3 (High: 0.08–0.10)

The findings for the effective connectivity analysis in the ISF3 are presented in Table 2 and Figure 4. The Kruskal–Wallis H test demonstrated no significant group-level differences in the effective connectivity measures in the ISF3 band at all three time points (T1, T2, T3). Thus, no Mann–Whitney U tests to individually compare the active treatment groups to the sham group were conducted.

### 3.2. Correlation with Clinical Measures

Table 3 presents the descriptive data of the mean differences for all the clinical measures at all timepoints. The findings of the correlation analyses between the effective connectivity measures and the clinical outcomes are presented in Table 4. A significant negative correlation was noted between the changes in effective connectivity in the direction of the pgACC to S1L and the changes in the pain severity (Table 4 and Figure 5). No significant correlations were noted between the other effective connectivity measures and the clinical measures.

## 4. Discussion

To our knowledge, our study is the first double-blinded randomised placebo-controlled trial to explore the effect of EEG ISF-NF on the effective connectivity between the targeted brain regions, and its associations with the changes in clinical outcomes of pain and disability, as these are important and recommended domains for determining the specificity of the EEG-NF training [11]. This study demonstrated a significant effect of ISF-NF up-training pgACC on the effective connectivity when compared to sham-NF training, and furthermore, the amount of information transfer from the pgACC to the S1L was correlated with the amount of pain severity reduction. While only a few previous studies have demonstrated objective neural changes following EEG-NF training [9,10,11], our study is the first study to demonstrate evidence of correlations of objective neural changes with improvements in clinical outcomes, thus indicating the specificity of the EEG-NF training.

EEG-NF for retraining the aberrant cortical activity in people with chronic pain has attracted significant interest over the past decade [9,10,11]. The reason for this is that neuromodulation techniques that are based on transcranial stimulation, either transcranial magnetic stimulation or transcranial electrical stimulation, merely disrupt activity and connectivity. However, it is unclear whether the brain really learns anything from these stimulations. From a theoretical perspective, a technique such as neurofeedback in which the brain learns how to function better via operant conditioning [1] may be preferable in the long run. A problem with neurofeedback is that recent systematic reviews highlight that the evidence for the effect of NF treatment for chronic pain, although promising, is of low quality, largely based on case-series and non-randomised studies [9,10,11]. Furthermore, studies investigating the effect of EEG-NF training for treatment of CLBP are lacking, with only one open-label study reported to date [56].

Chronic pain may be the consequence of a loss of pain inhibition rather than a persistent increase in pain input [2,3,23,24]. Indeed, chronic pain used to be defined as pain that extends beyond the period of healing of the original insult or injury, and hence lacks the acute warning function of physiological nociception [57]. The descending pain inhibitory pathway reflects the brain’s capacity to suppress acute or ongoing pain, and it can be assumed that a fully functioning system suppresses all pain, that a completely dysfunctional system results in constant pain, and a deficient system results in fluctuating pain [2]. Consequently, when the pgACC sends inhibitory information to the somatosensory cortex, this results in pain suppression [2,3,24]. Whereas acute pain usually results from more pain input, chronic pain likely results from a deficiency of pain inhibition [58]. The deficiency of the pgACC, as the main hub in the descending pain inhibitory system, triggers the development of chronic pain, well-known in one form of chronic widespread pain: fibromyalgia [23,59,60]. Yet, it has been suggested that this pgACC/vmPFC deficiency may underlie more forms of chronic pain [61]. Both structural [62] and functional [63] connectivity studies have indeed shown that pain chronification results from altered connectivity between the reward system and the pgACC, suggesting that the reward system maintains the deficiency of the pgACC in suppressing pain. Thus, based on the results of this secondary analysis, a neurofeedback protocol that strengthens the effective connectivity from the pgACC to the somatosensory cortex may be optimal. Yet, based on a study in spinal cord stimulation in which effective connectivity changes were computed [2,3], pgACC to dACC strengthening may also be of benefit.

We furthermore also observed a time course effect in neurophysiological measures in the EEG-NF up-training pgACC group, where the effective connectivity continued to increase over time. These findings have also been observed previously [64,65], where clinical symptoms and neurophysiological variables neither regressed to baseline nor remained stable but continued to improve for weeks following NF training. While this might be due to practice effects of learning to control neural activity or reflect slow consolidation processes [66,67], other mechanistic speculations such as the self-reinforcement of the brain over time to strengthen the correlational structure of network brain activity have also been proposed [64] and require further research. Based on these findings, it is recommended that future studies should include a longer follow-up period to sample the time point of greatest effect [64].

It is important to emphasize that this pilot study was not conducted and did not have power to determine the treatment effectiveness. Nevertheless, descriptive findings on clinical measures demonstrate a decreasing trend in pain and disability in all treatment groups. Our results are comparable to the previous NF studies in chronic pain conditions, who also demonstrated significant reductions in pain and disability following training [9,10,11]. We observed greater reductions in pain and disability in the pgACC group when compared to the other treatment groups. This was counter to our hypothesis. We hypothesized, based on theoretical underpinnings [2,3,23,24], that the rebalancing ratio training would be superior to simple pgACC up-training. The reason that pgACC training yielded the best results may be due to the simplicity of the pgACC protocol and ease in learning. Uptraining the current density of a single brain region may be easier for the brain to learn than learning how to self-regulate multiple areas, and certainly when they need to be trained in the opposite direction. Previous NF studies in chronic pain conditions have demonstrated that complex training protocols (e.g., adding beta down training to the protocol of uptraining of sensorimotor rhythm and down training of theta) reduced training effectiveness, but increasing the number of training sessions resulted in increased pain reduction for such protocols [9]. These findings suggest that training multiple cortical regions and more complex protocols (e.g., simultaneously uptraining and down training different brain regions) might require an increased number of training sessions.

We used sourced localised EEG-NF for training the cortical activity at the specific brain regions responsible for pain processing. Previous EEG-NF studies, however, have used conventional neurofeedback techniques, reinforcing alpha (~8–12 Hz) or sensorimotor (~12–15 Hz) rhythms and suppressing a combination of theta (~4–8 Hz) and/or beta (~13–30 Hz) rhythms, at one to two specific electrode levels [9,10,11]. However, only one study evaluated source-localized EEG-NF training of the ISF (<0.01 Hz) at three cortical regions simultaneously for the treatment of chronic pain [68]. The source-localized EEG-NF permits training specific and multiple brain regions simultaneously, potentially changing the activity in the trained area and connectivity to/from the trained area [69,70]. However, superiority to traditional sensor level neurofeedback training is still lacking [71].

This study has some limitations. The primary limitation of this pilot study is the small sample size, and a fully powered trial is required before drawing any definitive conclusions based on this study’s observations. Another limitation is that the XYZ coordinates for the three selected areas were based on a neurosynth meta-analysis of pain. While this should be optimal for the group, this may be suboptimal for the individual. For example, the selection of the pgACC XYZ coordinates was based on the neurosynth meta-analysis for pain, yet there may be more optimal ROIs to target for the individual patient. Similarly, the connectivity between the somatosensory cortex and salience network is topographic, i.e., based on which part of the body is in pain [5], and thus training the entire somatosensory cortex may be suboptimal in comparison to only training the XYZ coordinates that somatotopically relate to the painful body area. Lastly, this pilot study was a secondary analysis of a pilot randomised placebo-controlled study conducted to explore the feasibility and safety of a source-localized EEG-NF for the treatment of CLBP. Hence, the study did not have power to determine the treatment’s effectiveness. Future full powered trials are required to determine treatment effectiveness.

## 5. Conclusions

The ISF-NF up-training the pgACC can reduce pain experience via the influence the effective connectivity from pgACC to S1L. This pilot study provided estimates of changes in the effective connectivity measures and clinical outcomes following EEG-NF. These estimates will be used to determine the sample size and design a future fully powered trial. Further, this pilot secondary analysis also provides methodological insights for developing and testing the efficacy of a novel neurofeedback intervention to improve outcomes in people with CLBP. Future studies could test if specifically strengthening the effective connectivity from pgACC to S1L via brain–computer interface training could result in improved clinical outcomes.

## Figures and Tables

**Figure 1 brainsci-12-01514-f001:**
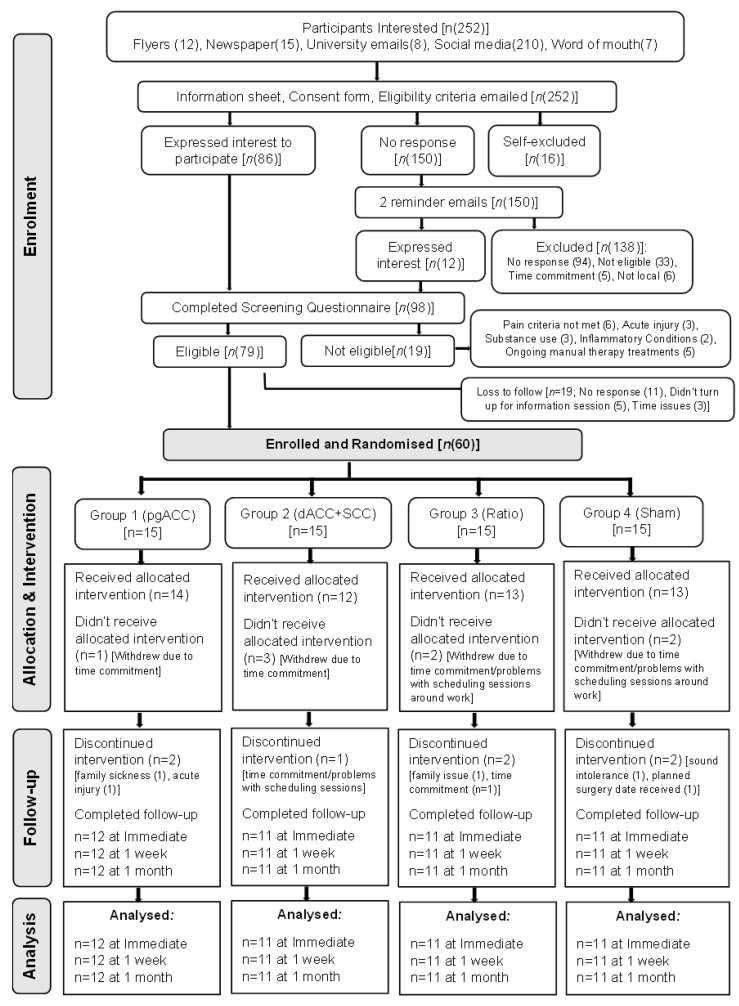
Flow of participants through the study phases. pgACC: pregenual anterior cingulate cortex; dACC: dorsal anterior cingulate cortex; SSC: primary somatosensory cortex (S1).

**Figure 2 brainsci-12-01514-f002:**
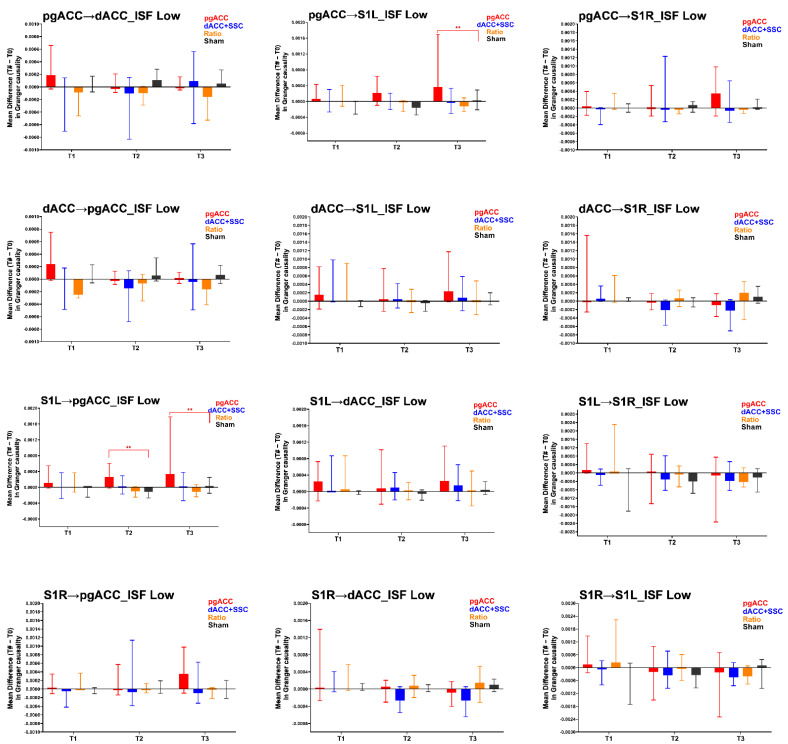
Results of the effective connectivity analysis for the ISF1 band (low: 0.01–0.04 Hz). pgACC: pregenual anterior cingulate cortex; dACC: dorsal anterior cingulate cortex; S1L: primary somatosensory cortex left; S1R: primary somatosensory cortex right; ISF: infraslow frequency; T1: changes immediately post-treatment when compared to baseline (T1–T0); T2: changes at one-week follow up when compared to baseline (T2–T0); T3: changes at one-month follow up when compared to baseline (T3–T0); ->: effective connectivity in the direction of one region to other. Higher values represent increase in the effective connectivity compared to baseline. **: *p* < 0.017.

**Figure 3 brainsci-12-01514-f003:**
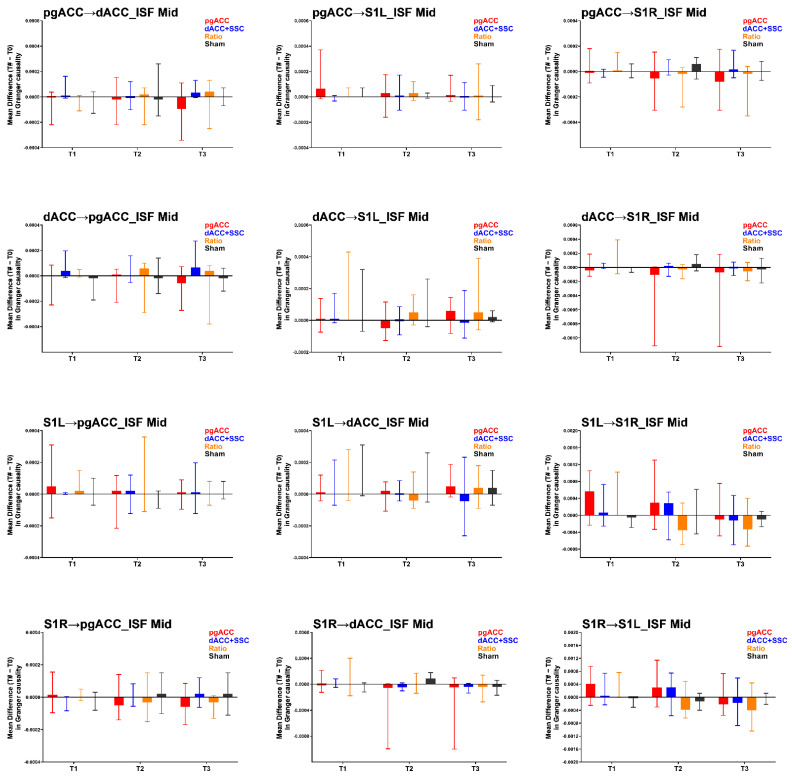
Results of the effective connectivity analysis for the ISF2 band (mid: 0.05–0.07 Hz). pgACC: pregenual anterior cingulate cortex; dACC: dorsal anterior cingulate cortex; S1L: primary somatosensory cortex left; S1R: primary somatosensory cortex right; ISF: infraslow frequency; T1: changes immediately post-treatment when compared to baseline (T1–T0); T2: changes at one-week follow up when compared to baseline (T2–T0); T3: changes at one-month follow up when compared to baseline (T3–T0); ->: effective connectivity in the direction of one region to other. Higher values represent increase in the effective connectivity compared to baseline.

**Figure 4 brainsci-12-01514-f004:**
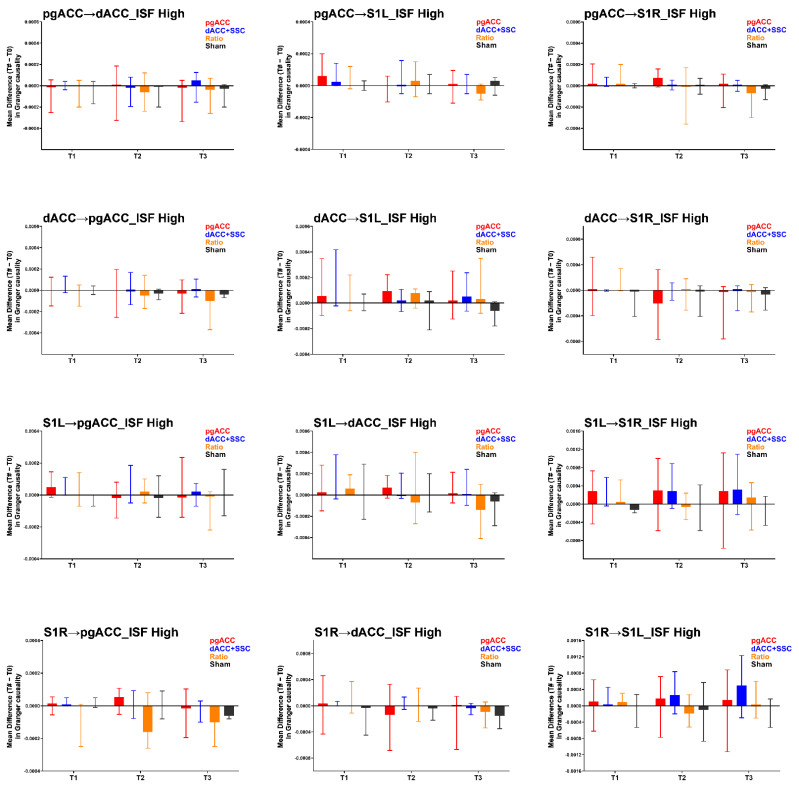
Results of the effective connectivity analysis for the ISF3 band (high: 0.08–0.1 Hz). pgACC: pregenual anterior cingulate cortex; dACC: dorsal anterior cingulate cortex; S1L: primary somatosensory cortex left; S1R: primary somatosensory cortex right; ISF: Infraslow frequency; T1: changes immediately post-treatment when compared to baseline (T1–T0); T2: changes at one-week follow up when compared to baseline (T2–T0); T3: changes at one-month follow up when compared to baseline (T3–T0); ->: effective connectivity in the direction of one region to other. Higher values represent increase in the effective connectivity compared to baseline.

**Figure 5 brainsci-12-01514-f005:**
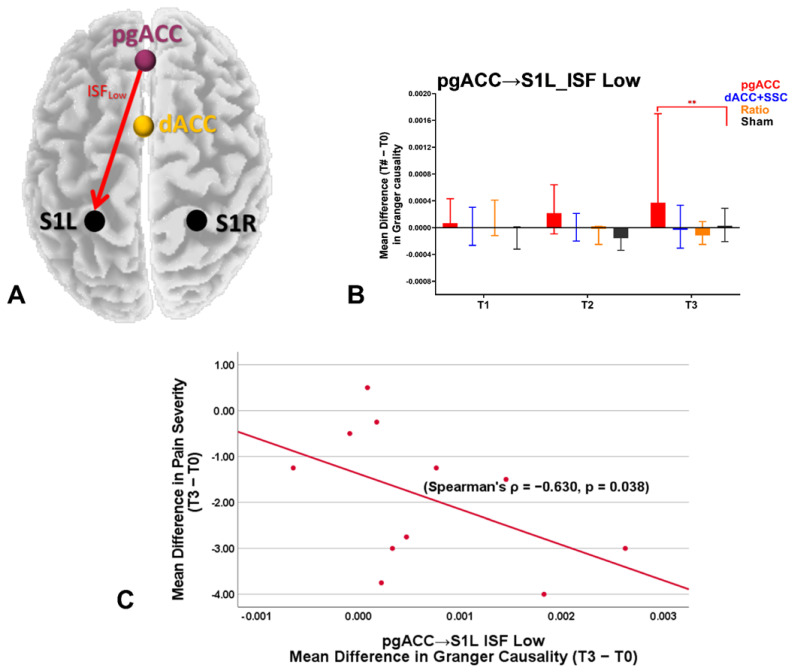
Summary of the results of the correlation analyses. In the ISF-NF up-training pgACC group, a significant increase in the effective connectivity was demonstrated in the direction of pgACC to the S1L (**A**), particularly in the ISF1 (low frequency) band at one-month follow-up (**B**), and the mean difference in the effective connectivity negatively correlated with the mean difference in the pain severity (**C**). **: Significant difference.

**Table 1 brainsci-12-01514-t001:** Demographics and clinical characteristics of participants.

Characteristics/ Measures	Group 1 (pgACC) (*n* = 12)	Group 2 (dACC + S1) (*n* = 11)	Group 3 (Ratio) (*n* = 11)	Group 4 (Sham) (*n* = 11)
Demographics				
Age (yrs) (Mean ± SD)	42.0 ± 15.2	41.4 ± 15.7	47.6 ± 14.0	44.2 ± 16.7
Sex				
Female; n (%)	7 (58)	9 (82)	7 (64)	8 (73)
Male; n (%)	5 (42)	2 (18)	4 (36)	3 (27)
Ethnicity				
NZ European; n (%)	9 (75)	5 (45)	10 (91)	6 (55)
Maori; n (%)	0 (0)	1 (9)	0 (0)	2 (18)
Indian; n (%)	0 (0)	1 (9)	0 (0)	0 (0)
Chinese; n (%)	1 (8)	1 (9)	0 (0)	1 (9)
Other; n (%)	2 (17)	3 (27)	1 (9)	2 (18)
Employment				
Employed; n (%)	7 (58)	4 (36)	7 (64)	5 (45)
Unemployed; n (%)	0 (0)	1 (9)	2 (18)	1 (9)
Retired; n (%)	0 (0)	2 (18)	1 (9)	1 (9)
Looking after family; n (%)	0 (0)	0 (0)	0 (0)	0 (0)
Self-employed; n (%)	2 (17)	1 (9)	0 (0)	3 (27)
Other; n (%)	3 (25)	3 (27)	1 (9)	1 (9)
Education				
University degree; n (%)	5 (42)	7 (64)	7 (64)	4 (36)
Trade/Apprenticeship; n (%)	3 (25)	1 (9)	1 (9)	3 (27)
Certificate/Diploma; n (%)	2 (17)	1 (9)	1 (9)	1 (9)
Year 12/equivalent; n (%)	2 (17)	1 (9)	0 (0)	2 (18)
Year 10/equivalent; n (%)	0 (0)	1 (9)	2 (18)	0 (0)
No formal qualification; n (%)	0 (0)	0 (0)	0 (0)	1 (9)
Pain variables				
Brief Pain Inventory				
Pain severity (Mean ± SD)	4.1 ± 2.0	3.7 ± 1.6	3.3 ± 1.4	3.7 ± 0.9
Pain interference (Mean ± SD)	4.3 ± 2.0	3.7 ± 1.7	4.5 ± 2.2	3.5 ± 1.8
Pain unpleasantness (Mean ± SD)	4.7 ± 2.6	4.0 ± 1.3	4.0 ± 1.7	4.4 ± 2.2
Pain bothersomeness (Mean ± SD)	4.8 ± 2.6	4.7 ± 1.8	3.9 ± 2.0	4.3 ± 2.4
Neuropathic pain (PainDetect) (Mean ± SD)	11.1 ± 6.3	12.7 ± 6.4	11.1 ± 7.8	9.7 ± 3.8
Central sensitisation (CSI) (Mean ± SD)	38.1 ± 13.5	36.6 ± 18.6	42.3 ± 8.4	36.5 ± 10.0
Functional status, quality of life, and sleep				
Disability (RMDQ) (Mean ± SD)	9.9 ± 3.6	9.7 ± 3.3	11.3 ± 5.8	9.4 ± 4.7
Well-being (WHO-5)	14.3 ± 4.0	13.6 ± 3.2	12.6 ± 5.0	15.3 ± 3.9
Quality of life (EQ-5D)				
Index score (Mean ± SD)	0.6 ± 0.3	0.8 ± 0.2	0.7 ± 0.2	0.8 ± 0.1
VAS (Mean ± SD)	74.1 ± 19.0	65.3 ± 17.5	61.4 ± 22.0	80.0 ± 11.4
Sleep (MOS-Sleep)				
Index I (Mean ± SD)	36.9 ± 21.5	36.4 ± 16.6	37.6 ± 13.7	33.3 ± 15.3
Index II (Mean ± SD)	41.1 ± 22.4	40.8 ±17.5	41.1 ± 13.9	36.8 ± 17.9
Psychological measures				
Pain catastrophising (PCS)				
Rumination (Mean ± SD)	5.5 ± 4.2	6.0 ± 3.1	5.3 ± 4.2	5.9 ± 4.2
Magnification (Mean ± SD)	3.2 ± 2.5	2.0 ± 1.8	2.6 ± 2.0	3.6 ± 2.7
Helplessness (Mean ± SD)	7.4 ± 4.3	5.4 ± 3.6	5.4 ± 2.7	7.6 ± 5.6
Total (Mean ± SD)	16.1 ± 10.0	13.4 ± 7.1	13.3 ± 6.7	17.1 ± 11.8
Pain vigilance and awareness (PVAQ) (Mean ± SD)	42.5 ± 13.0	43.1 ± 12.4	40.3 ± 9.0	40.7 ± 12.5
Depression (DASS-21) (Mean ± SD)	3.2 ± 2.4	3.4 ± 4.0	3.2 ± 2.4	3.9 ± 3.0
Anxiety (DASS-21) (Mean ± SD)	3.5 ± 3.7	3.6 ± 2.6	3.6 ± 4.2	3.5 ± 2.7
Stress (DASS-21) (Mean ± SD)	6.4 ± 4.5	5.7 ± 3.9	8.4 ± 3.7	7.5 ± 3.6
Positive Affect (PANAS) (Mean ± SD)	30.5 ± 5.8	31.6 ± 5.0	27.1 ± 6.1	29.9 ± 5.5
Negative Affect (PANAS) (Mean ± SD)	17.9 ± 6.5	16.7 ± 6.2	18.6 ± 6.0	19.4 ± 4.7
Illness perception (IPQ) (Mean ± SD)	49.1 ± 8.6	42.9 ± 5.4	44.5 ± 9.3	49.6 ± 13.6
Emotional regulation (ERQ)				
Cognitive reappraisal (Mean ± SD)	27.0 ± 7.5	31.5 ± 5.3	50.1 ± 7.7	28.3 ± 6.6
Emotional suppression (Mean ± SD)	14.7 ± 5.4	12.2 ± 4.2	28.5 ± 7.2	15.2 ± 4.6
Mindfulness (FFMQ) (Mean ± SD)	47.0 ± 8.0	51.7 ± 7.5	50.1 ± 7.7	47.9 ± 6.6
Treatment expectations				
Credibility (Mean ± SD)	17.6 ± 4.8	18.5 ± 4.4	18.5 ± 3.4	16.5 ± 4.7
Expectancy (%) (Mean ± SD)	56 ± 29	57 ± 27	44 ± 17	44 ± 25

pgACC: pregenual anterior cingulate cortex; dACC: dorsal anterior cingulate cortex; S1: Somatosensory cortex; CSI: Central Sensitization Inventory; RMDQ: Roland-Morris Disability Questionnaire; MOS: Medical Outcomes Study; DASS: Depression, Anxiety, and Stress Scale; PCS: Pain Catastrophizing Scale; PANAS: Positive and Negative Affect Schedule-short form; ERQ: Emotion Regulation Questionnaire; FFMQ: Five-Facet Mindfulness Questionnaire; EQ-5D: European Quality of Life; WHO-5: WHO-Five Well-Being Index; VAS: Visual Analogue Scale, IPQ: Illness Perception Questionnaire; SD: Standard deviations.

**Table 2 brainsci-12-01514-t002:** Between-group comparisons for the effective connectivity measures.

Effectivity Connectivity Measure	ISF1 (Kruskal–Wallis H, *p* Value)			ISF2 (Kruskal–Wallis H, *p* Value)			ISF3 (Kruskal–Wallis H, *p* Value)		
	T1	T2	T3	T1	T2	T3	T1	T2	T3
pgACC➔dACC	6.52, 0.089	5.10, 0.165	6.85, 0.077	2.36, 0.501	0.47, 0.925	3.43, 0.329	0.75, 0.861	0.20, 0.977	2.26, 0.519
pgACC➔S1L	3.19, 0.363	5.58, 0.134	8.29, 0.040 *	2.92, 0.404	0.798, 0.850	0.64, 0.887	3.63, 0.304	0.33, 0.955	1.75, 0.627
pgACC➔S1R	2.32, 0.509	1.71, 0.634	2.98, 0.395	0.82, 0.844	2.45, 0.485	2.33, 0.506	2.16, 0.539	1.90, 0.593	1.60, 0.660
dACC➔pgACC	7.30, 0.063	4.01, 0.260	5.06, 0.167	4.62, 0.202	0.60, 0.896	5.10, 0.165	1.36, 0.715	0.73, 0.866	3.75, 0.289
dACC➔S1L	2.80, 0.423	2.63, 0.452	4.27, 0.238	3.42, 0.332	2.40, 0.494	0.989, 0.804	1.46, 0.691	3.33, 0.343	3.43, 0.330
dACC➔S1R	1.22, 0.749	3.18, 0.365	3.91, 0.271	1.61, 0.658	3.99, 0.262	0.682, 0.877	2.20, 0.531	2.10, 0.553	0.432, 0.934
S1L➔pgACC	3.66, 0.301	7.70, 0.053 *	7.91, 0.048 *	2.50, 0.475	0.80, 0.849	0.03, 0.998	3.75, 0.290	1.03, 0.794	1.05, 0.790
S1L➔dACC	3.37, 0.337	2.37, 0.499	3.73, 0.292	0.16, 0.983	0.38, 0.944	0.854, 0.837	0.752, 0.861	1.91, 0.591	2.60, 0.458
S1L➔S1R	5.91, 0.423	2.80, 0.423	0.256, 0.968	2.08, 0.556	2.48, 0.480	1.29, 0.731	3.51, 0.320	2.32, 0.508	3.45, 0.328
S1R➔pgACC	4.70, 0.196	0.47, 0.926	2.85, 0.416	1.87, 0.600	1.23, 0.745	3.77, 0.287	1.51, 0.679	4.04, 0.257	5.11, 0.164
S1R➔dACC	0.83, 0.842	3.20, 0.362	4.10, 0.251	1.18, 0.758	5.88, 0.118	0.10, 0.992	4.50, 0.213	2.44, 0.486	0.91, 0.823
S1R➔S1L	5.26, 0.154	1.49, 0.685	0.29, 0961	1.72, 0.634	2.22, 0.529	0.994, 0.803	1.74, 0.628	2.64, 0.451	3.30, 0.348

*: Significant difference.

**Table 3 brainsci-12-01514-t003:** Descriptive data for the clinical measures at all timepoints.

Variable	Time Point	Group 1 (pgACC) (*n* = 12)	Group 2 (dACC + S1) (*n* = 11)	Group 3 (Ratio) (*n* = 11)	Group 4 (Sham) (*n* = 11)
BPI: Pain severity MD (95% CI)	T1–T0	−1.5 (−2.6, −0.3)	−1.3 (−2.9, 0.2)	−0.1 (−0.6, 0.4)	−0.5 (−1.4, 0.5)
	T2–T0	−1.8 (−2.6, −1.1)	−1.2 (−3.5, 1.1)	0.1 (−1.1, 1.3)	−1.1 (−1.6, −0.5)
	T3–T0	−1.9 (−2.9, −0.9)	−1.3 (−2.7, 0.1)	−0.3 (−1.3, 0.7)	−1.5 (−2.3, −0.6)
BP: Pain Interf. MD (95% CI)	T1–T0	−2.3 (−3.6, −1.1)	−1.0 (−2.2, 0.1)	−1.5 (−2.9, −0.2)	−0.3 (−1.1, 0.6)
	T2–T0	−2.3 (−3.4, −1.3)	−1.0 (−2.5, 0.5)	−1.5 (−2.6, −0.3)	−0.9 (−1.5, −0.3)
	T3–T0	−2.5 (−4.1, −1.0)	−0.9 (−2.7, 0.9)	−1.8 (−3.2, −0.5)	−1.6 (−2.5, −0.6)
Pain Unpleasantness MD (95% CI)	T1–T0	−2.2 (−3.5, −0.9)	−1.3 (−2.5, −0.1)	−0.3 (−1.9, 1.4)	0.2 (−1.1, 1.5)
	T2–T0	−1.6 (−2.7, −0.4)	−1.1 (−3.2, 1.0)	0.5 (−1.0, 1.9)	−1.0 (−2.2, 0.2)
	T3–T0	−2.2 (−3.6, −0.8)	−0.8 (−2.3, 0.7)	−0.5 (−1.9, 1.0)	−1.2 (−3.0, 0.6)
Pain Bothersomeness MD (95% CI)	T1–T0	−2.7 (−4.5, −0.9)	−1.9 (−3.1, −0.7)	−0.5 (−2.1, 1.2)	−0.6 (−1.8, 0.5)
	T2–T0	−2.5 (−4.1, −0.8)	−2.1 (−3.5, −0.7)	−0.4 (−2.3, 1.6)	−1.4 (−2.2, −0.5)
	T3–T0	−2.6 (−4.2, −0.9)	−1.0 (−2.7, 0.7)	−0.5 (−2.1, 1.2)	−1.6 (−2.9, −0.4)
RMDQ MD (95% CI)	T1–T0	−4.5 (−6.7, −2.2)	−2.8 (−5.6, 0.0)	−4.2 (−8.2, −0.2)	−1.7 (−2.8, −0.6)
	T2–T0	−4.1 (−6.5, −1.7)	−3.7 (−6.1, −1.3)	−4.5 (−8.6, −0.5)	−1.5 (−2.3, −0.6)
	T3–T0	−4.6 (−6.3, −3.0)	−4.0 (−6.6, −1.4)	−5.5 (−9.0, −1.9)	−2.2 (−3.2, −1.2)

BPI: Brief Pain Inventory; CI: confidence interval; dACC: dorsal anterior cingulate cortex; Intf.: interference; MD: mean difference; pgACC: pregenual anterior cingulate cortex; RMDQ: Roland–Morris Disability Questionnaire; S1: somatosensory cortex; SD: standard deviation; T0: baseline; T1: immediately post-treatment; T2: one-week follow up; T3: one-month follow up.

**Table 4 brainsci-12-01514-t004:** Result of the correlation analyses between the changes in the effective connectivity measures and the changes in clinical outcomes.

Clinical Variables (T#–T0)	ISF1 (Low)		
	pgACC➔S1L at One-Month Follow-Up (T3–T0) (Spearman’s Rho and *p* Value)	S1L➔pgACC at One-Week Follow-Up (T2–T0) (Spearman’s Rho and *p* Value)	S1L➔pgACC at One-Month Follow-Up (T3–T0) (Spearman’s Rho and *p* Value)
BPI_Pain severity	ρ = −0.630, *p* = 0.038 *	ρ = −0.406, *p* = 0.215	ρ = −0.507, *p* = 0.112
BPI_Pain interference	ρ = −0.114, *p* = 0.739	ρ = 0.105, *p* = 0.759	ρ = 0.059, *p* = 0.863
Pain unpleasantness	ρ = −0.554, *p* = 0.077	ρ = −0.248, *p* = 0.461	ρ = −0.402, *p* = 0.220
Pain bothersomeness	ρ = −0.487, *p* = 0.182	ρ = −0.326, *p* = 0.327	ρ = −0.437, *p* = 0.179
Disability (RMDQ)	ρ = 0.084, *p* = 0.805	ρ = −0.083, *p* = 0.808	ρ = 0.234, *p* = 0.488

* Indicates significant correlation. pgACC➔S1L: Granger causality from pregenual anterior cingulate cortex to left somatosensory cortex; S1L➔pgACC: Granger causality from left somatosensory cortex to pregenual anterior cingulate cortex; ISF: infraslow frequency; T0: at baseline; T2: at one-week follow-up; T3: at one-month follow up; BPI: Brief Pain Inventory; RMDQ: Roland–Morris Disability Questionnaire.

## Data Availability

Datasets generated and/or analysed during the current study are available from the corresponding author on reasonable request.

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
