# Peer review of "Infraslow Neurofeedback Training Alters Effective Connectivity in Individuals with Chronic Low Back Pain: A Secondary Analysis of a Pilot Randomized Placebo-Controlled Study"

_brainsci, 2022, doi:10.3390/brainsci12111514_

Round 1

Reviewer 1 Report

The authors investigated the This study explored the effect of electroencephalographic infraslow neurofeedback (EEG ISF-NF) training on effective connectivity and tested whether such effective connectivity changes are correlated with changes in pain and disability in people with chronic low back pain. The results are clear, and the data indicate that ISF-NF training pgACC can reduce pain via influencing effective connectivity between
pgACC to S1L. However, there are still some minor modifications and specific comments listed below.

·         The introduction should highlight the novelty of the paper. 

·         Please improve the discussion part. Comparisons are needed to justify the results.

·         Can the authors please elaborate on the significance of their findings to patients in future steps?

·         There are language errors that need to be corrected.

·         What is the limitation of this investigation?

Author Response

Comment 1:  The introduction should highlight the novelty of the paper. 

Response 1:  As this is a secondary analysis, the novel treatment approach used has been highlighted in the Study design section [Please see line 98-115].

Comment 2:  Please improve the discussion part. Comparisons are needed to justify the results.

Response 2:  The comparisons to previous studies has been included in the discussion part as suggested [Please see lines 390-393, 428-430, 441-443, and 459-464]

Comment 3:   Can the authors please elaborate on the significance of their findings to patients in future steps?

Response 3:  The significance of the findings has been added to the conclusion sections [Please see lines 487-491].

Comment 4:  There are language errors that need to be corrected.

Response 4:  The manuscript text has been reviewed again and all the language errors have been corrected.

Comment 5:  What is the limitation of this investigation?

Response 5:  The limitations of the study are listed in the last paragraph of the discussion. Further limitations have been added [Please see Lines 469-483].

Reviewer 2 Report

The authors present a study that explored the effect of infralent electroencephalographic neurofeedback (EEG ISF-NF) training on effective connectivity and tested whether such effective connectivity changes and are correlated with changes in pain and disability in people with chronic low back pain. The study is useful for obtaining neurofeedback and in creating Brain Computer Interface systems. The authors made a good presentation, well documented and argued. The presented results are relevant for the topic addressed.

I ask the authors to introduce in the Discussions chapter a comparison of the results obtained with other similar results mentioned in the specialized literature. By introducing this comparison, the article will have a better consistency.

Author Response

Comment 1: I ask the authors to introduce in the Discussions chapter a comparison of the results obtained with other similar results mentioned in the specialized literature. By introducing this comparison, the article will have a better consistency.

Response 1: The comparisons to previous studies have been included in the discussion part as suggested [Please see lines 390-393, 428-430, 441-443, and 459-464]